# Digital Twin Based SUDIHA Architecture to Smart Shopfloor Scheduling

**Hassan Khadiri** [1] (iD), **Souhail Sekkat** [2,*] (iD) **and Brahim Herrou** [1]

1   Laboratoire des Techniques Industrielles, FST USMBA, Fès BP 2202, Morocco;
    hassan_khadiri@yahoo.fr (H.K.); brahimherrou@yahoo.fr (B.H.)
2   Équipe MOSI-ENSAM—UMI, Meknès BP 15290, Morocco
*   Correspondence: s.sekkat@ensam.ac.ma; Tel.: +212-6-63300721

**Abstract:** Standing on the brink of the fourth industrial revolution, Cyber Physical Systems (CPS) are considered the basic components of the Smart Factory. One important challenge in cyber physical production systems is dynamic scheduling that can handle random disruptions such as failures, raw material shortages and quality defects. To achieve dynamic scheduling, we have proposed a Supervised and Distributed Holonic architecture we called SUDIHA. This architecture incorporates three Holons: Product Holon, Resource Holon and Order Holon and combines global supervision, achieved by Product Holon, with dynamic local control, achieved by Resource Holon. The Digital Twin (DT) concept is generally used to design CPS; it is virtual copies of the system that can interact with the physical counterparts in a bi-directional way. It seems to be promising to tackle the complexity and increase manufacturing system flexibility. In this paper, we use a DT Model to improve the SUDIHA architecture. We propose a Digital Twin based SUDIHA architecture (DT-SUDIHA). The paper will describe Digital Twins' configuration of each Holon of the SUDIHA Architecture, and the intelligent and real time data driven operation control of this architecture. A case study is carried out at the ENSAM-Meknes flexible workshop to prove the effectiveness of the proposed approach.

**Keywords:** digital twin; dynamic scheduling; multi-agent system; holonic manufacturing system; cyber physical systems; smart manufacturing

## 1. Introduction

In today's competitive markets, where mass customization is a new challenge, digitalization in manufacturing is seen as an opportunity to achieve higher levels of productivity [1]. Digitalization, also known as Industry 4.0, consists of using new technologies, including the Internet of Things (IoT), Augmented reality, Cyber Physical system (CPS), Cloud computing and analytics to revolutionize the way the products are manufactured and distributed [2]. A CPS is defined as a system that integrates a physical process and computational computer system. CPS technology [3] integrates sensing, computation and networking into physical objects to enable advanced shopfloor control and optimization. Smart factories leverage CPS technologies to create highly automated and connected production systems that can optimize production processes in real-time. In a smart factory, sensors collect data, which are analyzed using advanced analytics to identify areas for improvement and optimization. The result is a more efficient, effective and adaptable manufacturing system.

To better understand the concept of Cyber Physical Production Systems (CPPS), we will go back to the evolution of production systems' design paradigms. The first paradigm was the concept of Dedicated Transfer Lines, whose first goal was productivity. Integration in Manufacturing was the second paradigm based on the (CIM) approach, and aimed to achieve fully computerized and automated production systems integration. Reconfigurable Manufacturing Systems (RMS), proposed by Yoram Koren's team [4], is a paradigm that

used the Plug-and-produce concept to allow Manufacturing Systems components to be added and removed according to demand changes. These last years, with the development of Internet and Web technologies, Smart Manufacturing Systems (SMS) based on Cyber Physical Technology were proposed. SMS employs CPS, digital information technology and more flexible technical workforce training to achieve fast changes in production levels based on demand, and to optimize the supply chain [5].

One important challenge that must be met for Cyber Physical Production Systems Control is achieving automatic reactive scheduling, while the production is running, to face random disruptions such as delays, raw material shortages, quality defects and failures. CPPS involves the integration of physical and digital components to create autonomous systems that can interact with their environment and respond quickly to random disruptions. Scheduling autonomy in SMS is a critical capability that is enabled using advanced technologies, such as machine learning, IoT and DT. These technologies enable the system to adjust production schedules based on current conditions, such as changes in demand, equipment failures and other unexpected events. Autonomy is the ability to decide between a set of alternative tasks and to execute these tasks without human control [6]. To address the autonomy requirement and achieve dynamic scheduling, Holonic Architecture can be used [7]. We used, then, in the International Conference on Electrical, Computer, Communications and Mechatronics Engineering (ICECCME), an agent-oriented development process to propose a Supervised and Distributed Holonic architecture we called SUDIHA [8].

Industry 4.0 involves the integration of advanced technologies into the manufacturing process. Digitalization is a key aspect of I4.0, and it involves the use of digital technologies to optimize manufacturing processes. CPS are a key component of I4.0; they are made up of sensors, actuators and software that work together to monitor and control physical processes in real time [2]. By combining physical and digital systems, manufacturers can create more efficient and responsive production processes. One important CPS-based technology is the Digital Twin (DT). DTs provide a digital replica of a physical system, enabling real-time data collection, analysis and simulation of various scenarios [9]. The intelligence layer of a DT can be programmed with rules, algorithms and decision-making capabilities, allowing for more efficient scheduling tasks. DT technology has the potential to improve manufacturing operations management by enabling more accurate and timely decision-making, reducing downtime and costs and improving product quality. The implementation of DT-based distributed frameworks can further improve manufacturing operations management.

Using DTs to improve the scheduling function of CPPS is a promising topic. In this paper, we want to explore this topic. Therefore, we reviewed DT's application in manufacturing, and based on this review we developed a DT-based Holonic Architecture. Then, we carried out a case study at the ENSAM-Meknes flexible work cell to prove the effectiveness of the proposed approach. We propose, then, the design and implementation of a framework based on DTs for Cyber Physical Production System scheduling; it is an extension of our paper [8] where we proposed SUDIHA Supervised and Distributed Holonic architecture for dynamic scheduling. We will improve the SUDIHA architecture by using a DT Model and develop Digital Twin based SUDIHA architecture (DT-SUDIHA). This framework uses the Digital Twin concept to simulate the production system operation and, based on simulation results, propose a rescheduling action to improve efficiency. To validate the possibility to use DT-SUDIHA architecture in real industrial setups, we carry out a proof-of-concept in the ENSAM-Meknes flexible manufacturing cell. This paper is structured as follows. Section 2 presents a review of DT Technology, its use in manufacturing and a state of the art DT-based Architecture used for scheduling. Section 3 describes the Supervised and Distributed Holonic architecture SUDIHA proposed for dynamic scheduling and the agent-oriented development process used for its design. Section 4 describes the distributed DT-based SUDIHA architecture, Digital Twins' configuration of each Holon of the SUDIHA Architecture and the intelligent and real-time data-driven operation control of this Architecture. It also presents framework proof-of-concept in a use case. In Section 5, the

expected objectives behind the development of the DT-SUDIHA architecture are outlined and based on a use case, the limitations to Digital Twin implementation in manufacturing are highlighted. The conclusions are provided in Section 6.

## 2. DT Technology and Its Use in Manufacturing

A Digital Twin (DT) can be a digital replica of an object in the physical world. DT technology collects real world data and uses them to create simulations and predict how a process will perform. It has been widely applied in the manufacturing field [10]. In this section, a review of DT technology is presented. We discuss its use in manufacturing, achieve a state of the art DT-based Architecture used for scheduling and describe the five-dimensional DT model, which is a popular model used for DT design.

### 2.1. Digital Twin DT Concept

A Digital Twin is a virtual model of a physical object or system that uses data from sensors, machines and other sources to simulate the behavior and performance of the real-world object or system. It can integrate with other technologies such as IoT, AI and software analytics to enhance their capabilities. Kritzinger [11] distinguished between Digital Model, Digital Shadow and Digital Twin based on the level of interaction between the physical and virtual objects, and the purposes for which they are used.

1. The Digital Model is a purely digital representation of a physical object, used to create simulations and predict how the physical object will perform. It does not entail any interaction between physical and virtual objects.
2. The Digital Shadow is a digital replica of a physical object that receives data updates from the physical object but does not send data back. It includes the Digital Model and collects real time data to simulate the Digital Model. The Digital Shadow is used to analyze the performance of the physical object in real time, without interfering with its operation.
3. The Digital Twin is a digital replica of a physical object that enables bi-directional data flow between the real system and the virtual one. It includes the Digital Model and the Digital Shadow and uses the data collected in real time to optimize the performance of the physical asset and predict its behavior in different conditions.

The concept of a Digital Twin can be applied to various levels of the CIM (Computer-Integrated Manufacturing) pyramid. At the lowest level, a Digital Twin can represent a single sensor or actuator, allowing for real-time monitoring and control. This can be useful for detecting and correcting issues before they become more significant problems. For example, for controlling a yogurt filling system, Vogel-heuser [12] proposed a software agent that can react to a tank level sensor failure. This agent is a smart program of the tank cyber part. It detects the failure of the tank level sensor, relies on the flow sensors of the tank, computes the inflow and outflow of liquid, closes the tank fill valve when the tank is full and it can replace the tank level sensor and avoid an overflow. It can then be considered as the Digital Twin of this sensor.

At the device level, a Digital Twin can represent an individual machine or piece of equipment. By creating a Digital Twin of a device, it is possible to monitor its performance, identify potential issues and optimize its operation. At the higher levels of the CIM pyramid, a Digital Twin can represent an entire production facility or even an entire supply chain. By creating a Digital Twin of a production facility, it is possible to simulate different scenarios, test changes to the production process and optimize operations. The latest generation of CNC machines, also known as Machine Tools 4.0, are Cyber Physical Machine Tools (CPMT) [13]. These machines combine the traditional mechanical aspects of a CNC machine with digital technologies, such as data acquisition devices, embedded computing capabilities and a Machine Tool Cyber Twin (MTCT). The MTCT is a virtual representation of the physical machine tool that is created using real-time data from sensors and other devices. It provides the machine with the ability to analyze and optimize its performance in real time, allowing for greater efficiency, precision and flexibility in manu-

facturing operations. The (MTCT) consists, therefore, on an Information Model, a Database, Analytics and M2M Interfaces. The embedded computing capabilities of MTCT enable them to perform more complex tasks than traditional CNC machines, such as predictive maintenance, self-diagnosis and self-optimization. This can reduce downtime and increase productivity, as well as improve product quality and consistency. The smart HMIs of CPMT allow easier communication between the machine and the operator, making it easier to monitor and control the machine's operation. The integration of digital technologies into CNC machines has the potential to revolutionize manufacturing processes and improve a company's competitiveness. Now that the DT concept has been clarified, we will review its application in manufacturing in the following.

### 2.2. Application of DT Concept in Manufacturing

Digital Twin (DT) has been widely applied in the manufacturing field, especially in Prognostic and Health Management, Scheduling and Quality management [14]. Abadi et al. [15] proposed a new flexible and automated system based on Digital Twins (DTs), case-based reasoning (CBR) and Ontologies to achieve an optimal selection of production parameters for a given complex product. In this paragraph, we carry out a literature review to explore the application of the DT concept in shopfloor scheduling.

Several authors used the DT concept to improve CPPS Scheduling [16–18]. Ding et al. [19] tackled the problem of introducing Digital Twin technologies to the shopfloor to facilitate the real-time scheduling of CPPS. They defined a Digital Twin based Cyber Physical Production System (DT-CPPS) framework that achieved synchronization between the physical shopfloor and cyber shopfloor and developed several modules for autonomously controlling the operations of this framework. The proposed approach enables autonomous manufacturing.

Cimino et al. [20] achieved a literature review about DT applications in manufacturing that showed that the two major topics DT applications do not mention are: the integration of the proposed DT with the software control tools of the manufacturing system and the limited set of services offered in these DT applications. To overcome these missing implementation aspects, they proposed a use case of a DT implementation in the assembly line of the Politecnico di Milano School of Management. This assembly line is used to produce a prototypical mobile phone. It consists of seven stations and an automated conveyor used to transport the products from station to another. The assembly line is controlled by an MES software and uses OPC UA platform to achieve communication and interoperability requirement. Matlab and Simulink are used for data acquisition and modeling. A virtual twin of each station was built in Simulink. While the production is running, this virtual twin connects to the physical station and allows real-time synchronization of the simulation.

To plan production execution, Novak et al. [21] proposed an approach focused on the utilization of a production planner based on Planning Domain Definition Language (PDDL). A Digital Twin was used to achieve coordination of shopfloor devices on runtime. To validate the proposed solution, an Industry 4.0 Testbed was used. It included four robots, five workstations and a Transport System with six autonomous shuttles. All these devices were interconnected by OPC UA communication.

To sum-up, we can say that this review shows that DTs open new possibilities to predict, optimize and improve shopfloor scheduling, but there are still some limitations. Indeed, well-established DT design methods that allow users to create and connect physical objects with their virtual counterpart is still missing. We will, then, explore the DT Design Process in the following.

### 2.3. DT Design Process

Digital Twins consist of 2D or 3D visual models, analytical models and behavioral models . . . DT have to manage the physical asset; it then has a library of standard tasks that the physical asset can perform. Therefore, in addition to the various models mentioned above, it must also contain a set of control programs. To ensure that the virtual model accurately reflects the behavior of the physical system, it is essential to update the virtual model

with real-time data from the physical system. By continuously updating the virtual model with real-time data, we can simulate the physical system. DT must then use simulations based on real time data acquisition. On the other hand, Component DT models (sensor DTs or actuator DTs, for example) can be combined to create a composite DT Model (Device DT, for example).

An important topic to address is to find out how to design and connect physical objects with their virtual counterpart. Several authors addressed this subject [10,22–24]. Lui et al. [10] conducted a review of DT enabling technologies used in industrial applications. This review showed that DT concepts should rely on industry practice and raised the issue of data integration and complex phenomena modeling. Segovia et al. [23] explored the problem in several phases: from DT Specification to DT Development, through Architectural Design and DT Modeling. They reviewed Communication Protocols Platforms and Tools used for Data Synchronization during Digital Twin Development. In the first step of Digital Twin design, developers must gather the physics and operational data of a Physical Object (PO) or system in order to develop a mathematical model that simulates the original; this is Digital Twining. It consists of building a DT of a PO in the cyber world and establishing data channels for cyber–physical synchronization. They then ensure that the virtual model can receive feedback from the real-world object through sensors. This lets the digital version mimic what is happening with the physical asset in real time, which allows it to detect potential problems and to have an idea of the performances.

The next step in the DT building process is to implement it. Developers must use protocols, tools and standards to develop and synchronize the PO and its Virtual Counterpart at this step. The creation of high-fidelity virtual models is an essential step in building a Digital Twin, as it allows one to model the geometry, physical properties, behaviors and rules of the PO or system as closely as possible. To create a Virtual Object (VO) of a PO, several DT Models have been proposed in the literature. A first three-dimensional DT Model was proposed in 2003 by Grieves, from the University of Michigan, in his product life cycle management (PLM) course [25]. This DT model included three parts: (1) a physical entity, (2) a virtual entity and (3) a connection that transfers information between physical and virtual entities. Then, a second extended five-dimensional DT Model was proposed by Fei TAO [24]. This extended DT model (Figure 1) added data and service dimensions to the previous model. Indeed, nowadays, data are extremely important in modeling, because they contain useful information for the simulation, optimization and prediction of the behavior of the physical entity. Service is another important element in modeling. To solve the heterogeneity problem and ensure interoperability requirements, we must encapsulate DT functions in standardized services with user-friendly interfaces for easy and on-demand use. To meet these new requirements, the five-dimensional DT model was designed. This model is able to support data fusion and on demand use services [26]. To understand the components of the five-dimensional DT and their roles in creating and maintaining a functional and efficient system, we use an analogy by comparing the five-dimensional DT to a body and its components to bodily organs. In this analogy, the Physical Entity (PE) can be compared with the skeleton, providing a physical structure and framework for the DT. The Virtual Entity (VE) can be compared with the heart, which pumps results and strategies to other components of the DT, much like the heart pumps blood to different parts of the body.

The Services (Ss) can be compared with the sense organs, such as the eyes or ears, which interact with users directly, collecting information and feedback. The DT Data (DD) can be compared with blood, which feeds the DT with valuable information continuously, just as blood provides nutrients and oxygen to the body's organs. The Connection (CN) can be compared with the blood vessels, carrying the data to different components of the DT, much like blood vessels transport blood to various parts of the body.

We can finally say that implementing a DT-based distributed framework can improve manufacturing operations management by enabling more accurate and timely decision-making. In this paper, we propose a framework based on DTs for smart scheduling in

Cyber Physical Production Systems. We develop Digital Twin based SUDIHA architecture (DT-SUDIHA). However, before describing the DT-based architecture we propose, we first review the SUDIHA architecture we have proposed for dynamic scheduling of CPPS [8].

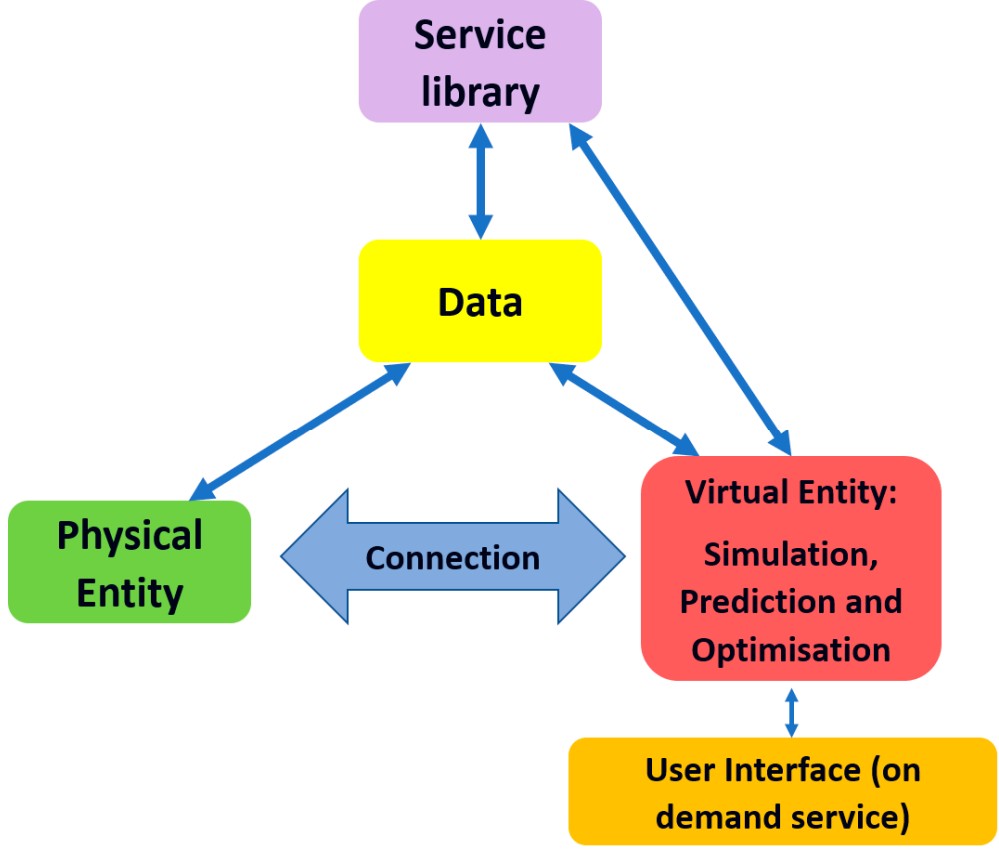

**Figure 1.** Five-dimensional DT concept model "Adapted from [24]".

## 3. SUDIHA Proposed Holonic Architecture

To achieve the scheduling of Cyber Physical Production Systems and to face random disruptions such as delays or failures, we have proposed a Holonic architecture that uses the OPT and Kanban methods. We called this control architecture SUDIHA, from SUpervised DIstributed Holonic architecture. In this paper, we improve the SUDIHA architecture by using a DT Model and we develop DT-based SUDIHA architecture for the dynamic scheduling of CPPS. So first, in this section, we describe SUDIHA architecture.

We used distributed scheduling and proposed an architecture that includes both a Supervised part and a Distributed part. On the one hand, in the Supervised part, a supervisory agent used the Optimized Production Technology OPT method to identify a bottleneck station, such as example station C in the case of Figure 2, and on the other hand, in the distributed part it used the Kanban method between the bottleneck and the upstream station and the priority rules in the other stations [8]. Since Holonic Architecture is relevant to achieve distributed control [7], we proposed it to achieve shopfloor scheduling and adopted the Multi-Agent methodology to develop this Holonic architecture [27,28]. Indeed, several Holonic Architectures were developed in manufacturing, among which we can mention PROSA (Product, Resource, Order, Staff architecture) [29], PROSIS (Product, Resource, Order, Simulation for Isoarchy Structure) [30], HCBA (Holonic Component Based architecture) [31], H2CM (Holonic Hybrid Control Model) [32] and ADACOR (ADAptive holonic COntrol aRchitecture) [33].

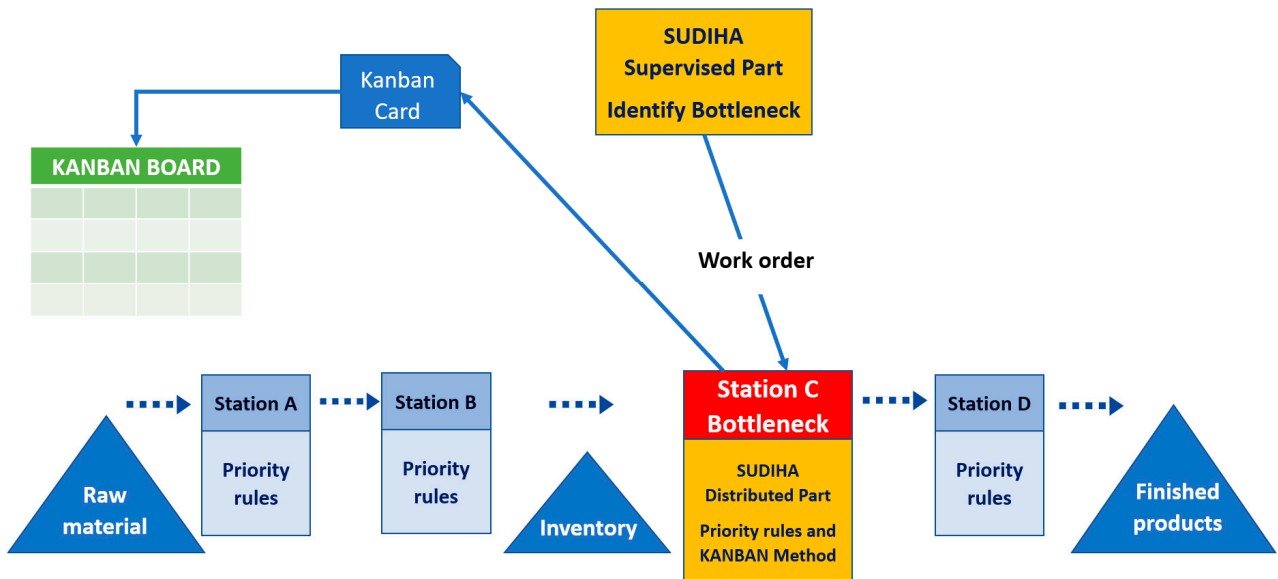

**Figure 2.** Proposed Multi-Agent Scheduling Architecture "Adapted from [34]".

SUDIHA architecture incorporates three Holons: Product Holon, Resource Holon and Order Holon. It provides a flexible and responsive approach to manufacturing by combining global supervision, achieved by Product Holon, with dynamic local control, achieved by Resource Holon.

Order Holon starts by identifying the bottleneck station and then sends work orders to the workshop. Product Holon, then supervises production and performs operations scheduling based on process plan constraints. Next, Resource Holon uses priority rules (FIFO, SPT, etc.) to control stations in a distributed way (Figure 3). Finally, a ConWip method is used at the bottleneck station, it returns Kanban Card to Order Holon and closes the loop. The use of holarchy and System of Systems SoS concept [35] in SUDIHA architecture allows for the creation of a more complex system from simpler ones. The high-level Order Holons can launch a production order at the cell level of the CIM Pyramid, while the low-level Order Holons carry out tasks at the device level. The agent-oriented development process was used to identify the basic functions of the multi-agent system and determine the types of agents required in the system and how they would interact. This process involves developing use cases to define the requirements and specifications of the system and then using these to design and develop the agents. There are several agent-oriented methodologies proposed in the literature, namely Gaia methodology [36], Tropos [37], MaSE methodology [38] and PASSI [39]. We used PROMETHEUS Methodology [40] and followed its steps, namely System specification and Architectural design, to develop SUDIHA architecture. We used also the advice provided by Detailed design, and Implementation steps of the methodology to update the scheduling software component of the ENSAM flexible manufacturing cell. We assume that a Holon is a set of programs grouped in a module and that all these programs achieve the same function or concern. Therefore, an important problem to solve is to define where all these Holons should be implemented.

Therefore, we will take a close look at the software tools used in shopfloor scheduling within a company. By running MRP computation and creating planned work orders, an ERP Software can achieve shopfloor scheduling. However, the operational plan or schedule proposed is static, and it cannot react to random disruptions. To achieve dynamic scheduling, SCADA software can be used. Indeed, in this software tool we can program and run a process plan of a product. However, this solution can only be used if the product diversity and the workshop complexity is low. For more complex production environments with high product diversity and multiple machines, an MES (Manufacturing Execution

System) software is used to manage the entire production process, from transforming suggested work orders from an ERP (Enterprise Resource Planning) software into closed work orders for each shopfloor resource, to sending production instructions to workstations and tracking progress in real-time.

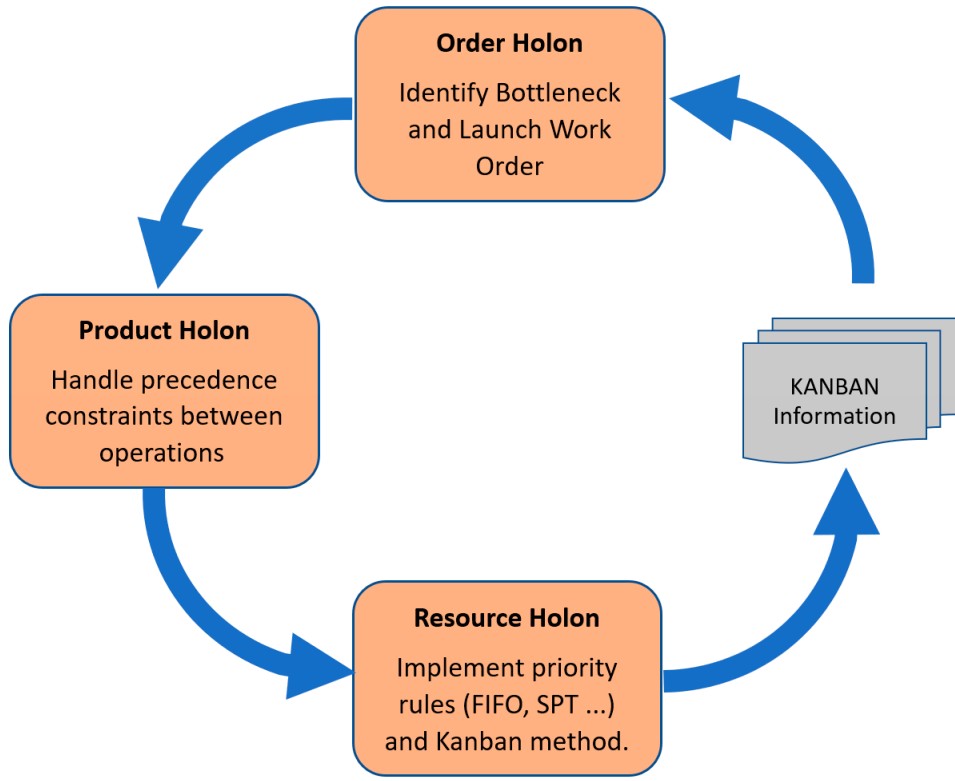

**Figure 3.** SUDIHA SUpervised DIstributed Holonic Architecture.

The Order Holon must be implemented in the ERP or MES software. It can use ODBC Driver to sends production jobs to the Product Holon. The Product Holon must be implemented in a SCADA software. It asks each Resource Holon to perform an operation. The Resource Holon control can be implemented at two levels: the High-Level Control (HLC), which allows decision making and achieving device coordination to be implemented in SCADA software, while the Low-Level Control (LLC), which manages physical equipment and allows carrying out actions requested by the HLC level, can be implemented in the Devices Control Units (DCUs). DCUs can be made of Computer Numeric Controls (CNCs) or Programmable Logic Controllers (PLCs) according to the hardware architecture of the DCU. HLC and LLC communicate, generally through OPC or TCP-IP drivers. Indeed, industrial automation providers are embracing the connectivity requirements of the digital factory and providing unique connectivity and data access services such as Rockwell's integrated OPC-UA servers [41].

To design SUDIHA architecture Holons, namely Order, Product and Resource Holons, we used an Agent Model developed by Wooldridge [42] and Russel [43]. An agent perceives its environment through sensors and acts upon that environment through actuators. The agent model includes a representation of the environment, which is updated based on the agent's perception. In addition to the environmental model, the agent also requires a definition of its goals. The agent program involves reasoning based on a knowledge base to select better actions. The knowledge base contains both reactive and proactive strategies for achieving its goals. The structure of Order Holon includes a BDI Belief-Desire-Intention model, a decision maker and a communication manager (see Figure 4). We were inspired by the model of the Product Agent architecture proposed by Kovalenko [44] and used the

BDI model proposed by Phung [45]. In [8] we proposed a sequence diagram showing the interaction between the three Holons of SUDIHA architecture.

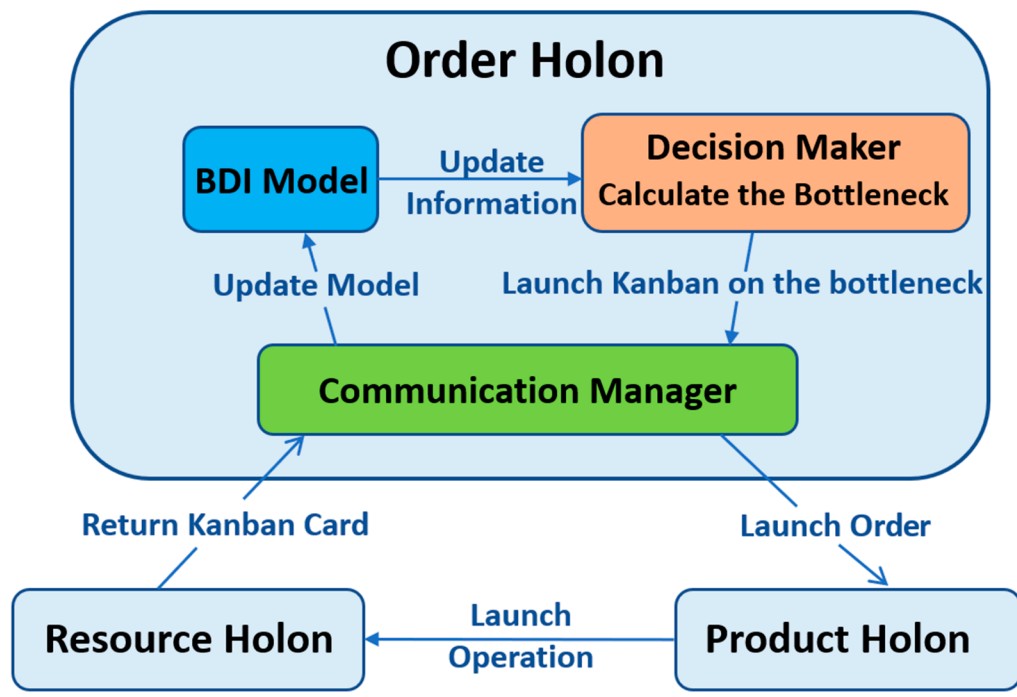

**Figure 4.** Structure of Order Holon.

The decision module of Order Holon determines, for each machine $M_j$, the total production time of all the jobs $T(M_j)$ as a function of $t_i(M_j)$ processing time of a job $Ji$ on a machine $M_j$, to find the bottleneck machine:

$$T(M_j) = \sum_{i=1}^{i=n} t_i(M_j) \tag{1}$$

In a second step, Product Holon launches an operation within the Resource Holon. Resource Holon defines a schedule by using priority rules (FIFO, SPT . . . ) to minimize machine vacancy time. Each Resource Holon is provided with a library of standard tasks it can run. The scheduling solution is the result of Holon interactions. This interaction consists of a message exchange between Holons.

Product Holons form a queue upstream of each resource. During the scheduling process, the ConWip method is implemented between the resource Holon and Order Holon. Each time a workpiece is processed in the bottleneck resource, its Holon returns a Kanban Card to Order Holon to release another Product Holon and close the loop. It then consults his message queue to know the workpiece it must perform.

Finally, we can conclude that the proposed scheduling architecture allows handling, in real time, random disruptions and is suitable for developing CPPS. We used the proposed approach to improve a scheduling software component of the ENSAM manufacturing cell. Indeed, we updated its software components so that it can identify the bottleneck and launch one by one work orders. In the following, we will improve the SUDIHA architecture by using the DT Model and we will develop DT-based SUDIHA architecture for dynamic scheduling of CPPS. DT is promising technology that can improve further dynamic scheduling and increase manufacturing system flexibility and autonomy.

## 4. DT-SUDIHA: Digital Twin Based SUDIHA Architecture

A production system combines the hardware, software and human resources of an organization into value added products in a controlled manner as per the policies of the

organization. An SMS should be more efficient and autonomous. Digital Twin can be used to achieve this autonomy. In a DT-based architecture, smart objects are defined as physical entities that are equipped with various types of sensors, actuators, and communication technologies that enable them to collect data about their environment, communicate with other devices and make decisions based on that data. These smart objects can range from simple sensors to complex systems such as robots or CNC Machine Tools. Smart objects play a critical role in a DT architecture, allowing for real-time monitoring, control and optimization. By providing intelligence and autonomy to physical entities, smart objects can help organizations achieve greater efficiency and productivity. We use the DT concept to improve the SUDIHA architecture. We will then design a digital object and establish data channel communication between the physical and virtual object to build DT-based SUDIHA architecture. In the following, we will present the DT-Based SUDIHA Framework we propose and describe the operation control flow we use in DT SUDIHA architecture to control the shopfloor from order to finished product delivery.

### 4.1. DT-Based SUDIHA Framework

In addition to their physical capabilities, smart objects in a DT architecture are also endowed with intelligence and autonomy. This means that they can process the data they collect and make decisions based on it. For example, a smart part equipped with RFID technology might be able to automatically identify itself and communicate its status to other devices, while a smart resource including, in addition to its Low-Level Controller (LLC), a High-Level Controller (HLC) might be able to carry out sophisticated reasoning and decision-making tasks.

To build DT-based SUDIHA architecture, we will start by defining a framework for this architecture. A DT-based Architecture typically consists of two components: the Physical Shopfloor (PS) and the Cyber Shopfloor (CS). The PS refers to the physical devices used in manufacturing, such as assembly lines, conveyor belts and robotic arms. The CS, which is the Digital Twin of the PS, allows for the optimization and simulation of manufacturing processes, and can be used to identify and correct any issues or inefficiencies in the PS. In DT-based Architecture, the virtual model can communicate with and control the physical equipment, and vice versa. This real-time communication and coordination improve PS flexibility and autonomy. Each Holon of DT-SUDIHA will consist of a physical part and a virtual part. It will therefore include three physical Holons: the smart product, the smart Resource and the smart Kanban. For each physical Holon, we will next provide its virtual twin: the virtual smart product, the virtual smart resource and the virtual smart order (Figure 5).

These Holons must be autonomous, and they must collaborate and exchange data during the production process. This communication is supported by the Digital Twin computation technologies in SUDIHA architecture. In first step of the Digital Twin Design Process, we will perform Twining by building DTs of Physical Objects in the Cyber world and establishing data channels for Cyber Physical synchronization. First, the Physical Shopfloor (PS) and the Cyber Shopfloor (CS) must be configured. Products, Resources and Orders must be equipped with an intelligence that allows them to proactively interact with each other in the PS to make decisions. Smart products must be able to know their process plans and the type of machine needed to perform each operation. They must also monitor their progress throughout the production process. Therefore, to make them smart, we attach an RFID tag to each product or batch; this way, the system can keep track of the sequence of manufacturing operations required and the type of machine needed to perform each operation. The RFID Reader installed on the stations of the shopfloor or on the device that transports the product can then send a notification about product arrival to a workstation. This reader is generally connected to SCADA Software through PLC Controller. Each product identity can be bound to a software entity that includes programs performing services requested by order Holons and a data structure that tracks the product during production. This software entity serves as the virtual counterpart of the product

and enables the system to monitor the progress of the product as it moves through the production process.

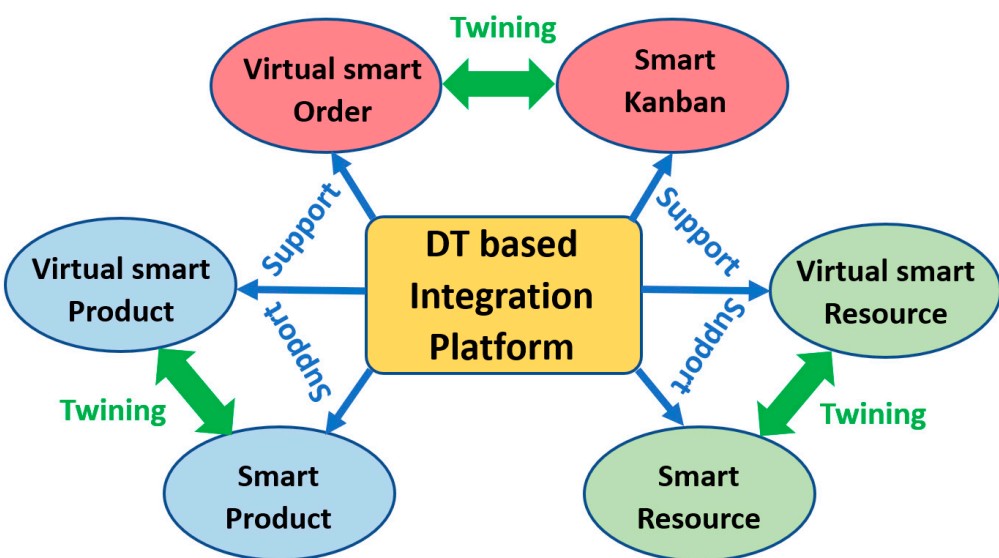

**Figure 5.** DT-Based SUDIHA Framework.

Resources such as machines, vehicles, and tools must also be smart and react to disturbance. Indeed, in addition to its Low-Level Control (LLC) implemented in the Device Control Units (DCUs) that manage generally physical Resources, we must endow its control system with High-Level Control (HLC). This level allows decision making and achieving device coordination. It is generally implemented in SCADA software; it is the Virtual counterpart of the Resource. Moreover, the reasoning is carried out in HLC of the Resource and depends on real-time data collected by LLC.

We use, then, the five dimensions of the DT model to combine the speed of LLC level and the reasoning carried out at HLC level to make the right decisions as quickly as possible and therefore meet the Cyber Physical Systems challenge. In addition to Data structure, which stores DT Data, each physical entity of a Resource is provided with a library of standard tasks it can run (services) (see Figure 6). To achieve data communication between Virtual and Physical parts, we use global variable or tags. Both Device and SCADA Software have access to these tags and can read and write on them. Orders materialized by Kanban Cards must also be smart. When a work order is launched, we follow its progress by associating it with the raw material that will be used to manufacture the final part. PS/CS configuration, or Twinning, will allow smart parts' and smart resources' cooperation during production to find, dynamically, an optimal production schedule and to react to disturbances.

*4.2. Operation Control Flow in DT SUDIHA Architecture*

Digital Twin technology allows real-time communication and coordination between PS and CS and better flexibility and autonomy of Shopfloor Control. We will, then, propose, in this paragraph, operation control flow of the part manufacturing lifecycle, based on the DT concept. This flow will start from work order and will go as far as product delivery. We will describe the Control flow of DT-SUDIHA architecture by a diagram (Figure 7) that follows the part manufacturing lifecycle from work order reception until finished part delivery.

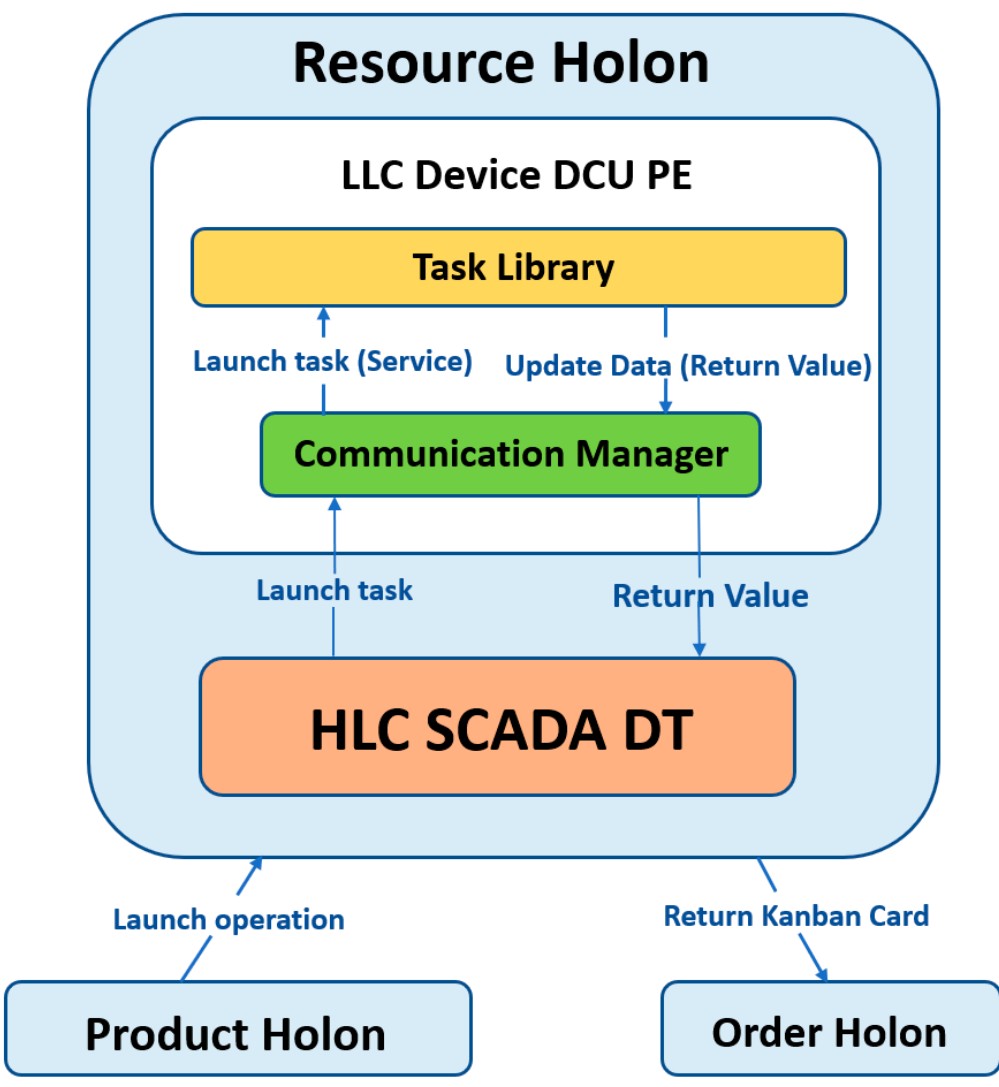

**Figure 6.** Structure of Resource Holon.

First, the user sends a product work order to the Order Holon; the latter must identify the bottleneck and decide, based on its state, when it can launch the manufacturing of this product. When this station is available, the Order Holon launches it on a Product Holon. Second, the Product Holon, based on the detailed information about the part, type, costs, process plans and the quantity, will generate the manufacturing operations for each entity of the part; then, it will allocate the manufacturing resources. A manufacturing operations plan will be simulated and, based on the simulation results, the Digital Twin model of the shopfloor will be updated.

If the simulation detects a disruption overstock, delays or unoccupied workstations, a scheduling module will be invoked to find an available solution. This scheduling module proposes an optimal plan by applying intelligent algorithms: Genetic Algorithm (GA), Artificial Neural Network (ANN) or Ant Colony Optimization (ACO). The planning proposed will undergo a follow-up simulation. In a third step, if the simulation result is OK, Product Holon launches an operation within the Resource Holon. Products form a queue upstream of each resource. Resource Holon defines a schedule by using priority rules (FIFO, SPT . . . ) to minimize machine vacancy time. During the scheduling process, the ConWip method is implemented between the resource Holon and Order Holon. Each time a workpiece is processed in the bottleneck resource, its Holon sends a message to Order Holon to release another Product Holon. It then consults his message queue to know the workpiece it must perform. In addition to the three, Holons of the SUDIHA architecture, the new control

system developed according to DT-SUDIHA architecture will have to include two other additional modules: namely, Simulation Module and Reactive Scheduling Module. On the other hand, an RFID tag attached to the product or product carrier allows the virtual product to be synchronized with the real product, and a High-Level Control (HLC) allows virtual resources to be synchronized with the real state of the physical resources. The Simulation Module and the Scheduling Module are based heavily on the real-time data collected by these devices.

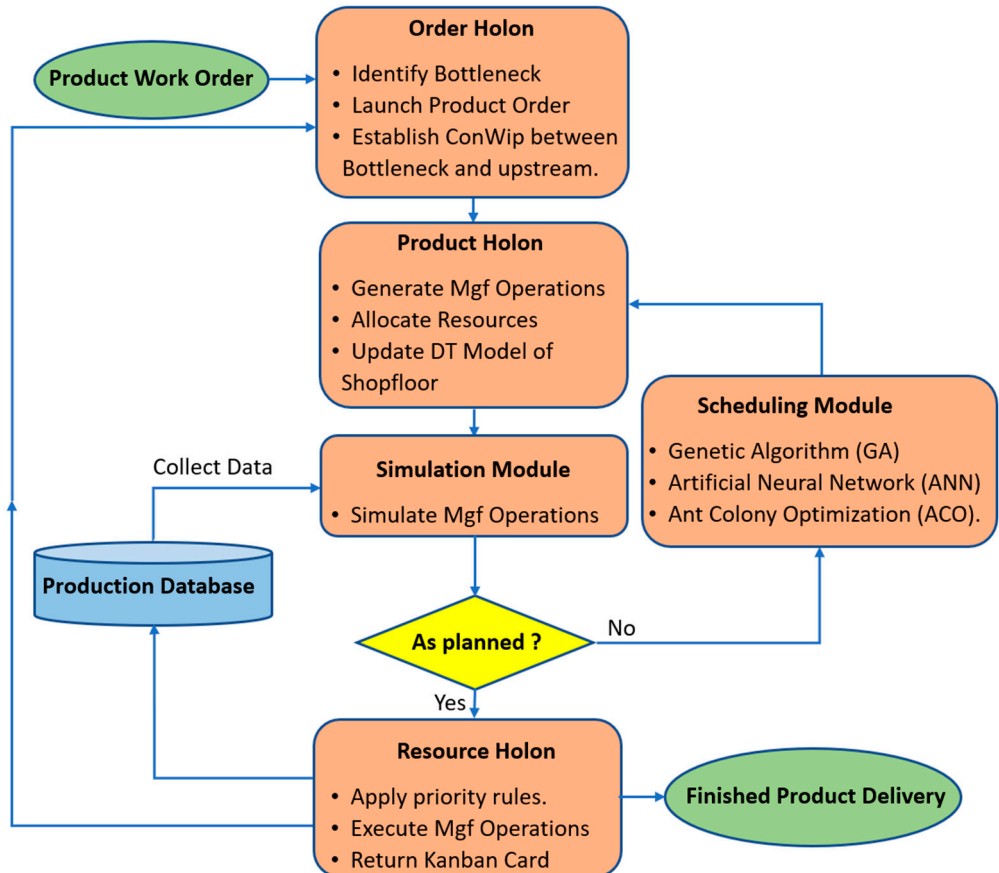

**Figure 7.** Control flow of DT-SUDIHA Architecture.

### 4.3. Case Study

The DT-SUDIHA architecture was applied to the flexible machining cell at the High School of engineering ENSAM. We used it to improve the scheduling software used for its control. As it was said, we used Holonic architecture to improve the scheduling of this machining cell; we will improve it further using the DT concept. The results of real-time simulation will be used to achieve a closed loop cell control. In the following, we will describe the ENSAM machining cell and Digital twining of the cell entities and the modules we have added to improve the cell scheduling.

The Manufacturing Cell included a Pallet system conveyor, Automatic Storage and Retrieval System, a CNC milling machine equipped with fives axes robots and an assembly station equipped with six axes robots, as shown in Figure 8. Control architecture of the system included SCADA software (CIROS Supervision). A network connected station controllers to the SCADA. The shopfloor control was achieved by an MES software (ICIM Production Manager). To implement a DT-based scheduling approach, CIROS Supervision [46] and CIROS Simulation 2008 version [47] software were used. An Integration Platform (IP) was developed on CIROS to allow communication between the components of the cell. This platform included several software modules; the most important were:

- Prod_OrderList(), which read the work order and saved it into global variables.
- Prod_Strategie(), which launched the operations of this WO on the Devices.
- ExecProg(), associated with each machine dealing locally with the resource constraints.

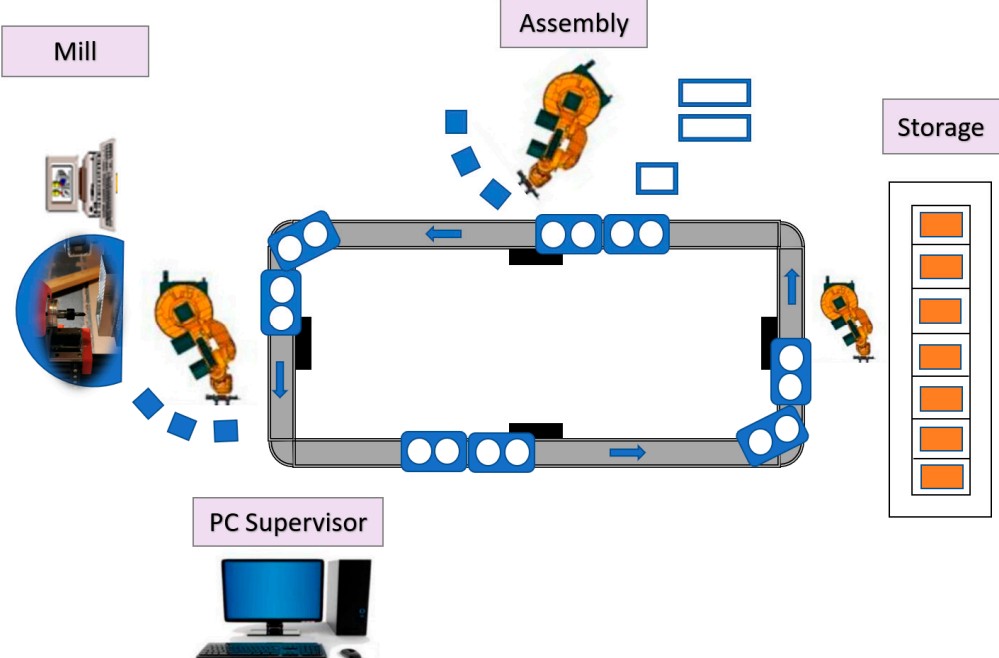

**Figure 8.** ENSAM flexible manufacturing cell.

The scheduling architecture was only valid if the cell included only a small number of workstations and if the number of work orders launched was low. Otherwise, the scheduling programs led to a chaotic situation with overloaded inventory at the bottleneck workstation. We used SUDIHA architecture to develop a smart control tool for the work cell. As it was said, we, in a first work, updated the Prod_OrderList() program to develop Order Holon that identified the bottleneck and launches work orders one by one. Then, locally, we updated the ExecProg() program to implement the Kanban Method and priority rules.

To improve further the scheduling of the cell, we updated the control system according to DT-SUDIDA architecture. In addition to the three Holons of the Manufacturing Cell Integration Platform, we developed two other additional modules: namely, Simulation Module and Reactive Scheduling Module. Each time a workpiece was processed or a disruption occurred, the scheduling module was invoked, which applied a Genetic Algorithm to propose a scheduling plan. This plan was simulated in CIROS Simulation before being sent to the Resource Holon for execution.

On the other hand, we used the RFID tag to build the Product Twin. Indeed, in the ENSAM machining cell, an RFID tag was fitted to the Carrier that transports the product and an RFID Reader was installed on the Transport system. This reader was connected to SCADA Software through Transport Controller.

Each Carrier transporting the product was bound to the processes that performed the services requested by Order Holon. The Product Identification was associated with a data structure that tracks the state of the product during production. The control system used the information provided by Part RFID to know the process plan it would use to produce it. Indeed, when a part X arrives at a workstation, the SCADA system, based on the part RFID, knows the operation it must undergo and the robot program it must use to grip it at this station.

We also built Station Twins in High-Level Control (HLC) of the station. These twins are designed to perform tasks requested by Product Holons. For example, the Transport Twin was implemented in CIROS Supervision; it includes data structure and program.

When a Product Holon sent a task to the Transport System, the Transport Twin program allocated this task to a suitable Carrier and sent service start messages to this Carrier.

To execute this service, the Carrier requested actuation by sending request messages to Low-Level Control (LLC) of the Transport through the communication manager. The data structure saved the Transport Carrier's status. These data were updated each time a Carrier arrived at a station. LLC of the Transport was implemented in the PLC. Once the requested actuation was performed by the PLC, the LLC replied with a confirmation and updated the Transport Status.

## 5. Discussion

Digital Twin technologies can be used to enable Smart manufacturing paradigms. DT-SUDIHA is a theoretical approach to apply DT technology to make production systems smarter. To achieve this result, two main challenges must be overcome.

The first one is establishing data channels between the Cyber Shopfloor (CS) and the Physical Shopfloor (PS). Real-time data collected from PS need to be transmitted to CS for simulation and optimization. This requires the use of data integration and processing software such as SCADA (Supervisory Control and Data Acquisition) systems. SCADA Software is used to collect real time data and transform it into information to make decision. The real time data collected are also used for simulation. After decision making, a follow-up simulation is used to ensure that the goals of smart manufacturing systems, such as improved efficiency, flexibility and autonomy, are achieved.

The second challenge is interoperability. This consists of the ability to exchange data among heterogeneous resources and product identifiers. To overcome this challenge, it is important to establish uniform data protocols and interfaces. MQTT (Message Queuing Telemetry Transport) and MTConnect are some of the preliminary outcomes that have been developed to address this challenge.

DT-SUDIHA architecture's implementation in today's shopfloors is a way to reach the goals of SMS. It improves the flexibility, efficiency and autonomy of manufacturing operations. Indeed, if we compare this architecture to the architectures based on the DT proposed in the literature, we find that these latter are generally limited in the number of services offered, and they are usually not integrated with the existing control system. The DT proposed by Cimino et al., for example, is focused on energy consumption computation, does not reach the full DT potential and is closer to a Digital Shadow. The DT proposed by Novak et al. is focused on Production planning and scheduling. In the architecture proposed by Ding et al., an embedded system device equipped with a software application tool must be integrated to both the product and the resource to allow manufacturing data computing, storing and transmission. However, in a company, the resources, and especially the products, are not always embedded systems. The products are equipped with an RFID tag that can only achieve data acquisition. Data computing, storing and transmission must be achieved by another system. The DT-SUDIHA proposed architecture can be used to extract the data from any available server and use case used for its validation, which seems to be a good test bed to experimentally understand how the DT should be integrated inside any control system. However, there are currently limitations to the implementation of this approach.

One of the primary limitations is the lack of fidelity and accuracy of current simulation models. Indeed, the simulation is an approximate reflection of reality, and if data from the physical entity have not been collected or have been collected in the wrong way, the simulation gives bad results. Furthermore, Digital Twins rely on accurate and detailed models of physical systems, but these models can be difficult to create and maintain. Additionally, the fidelity of these models can be limited by the availability of data and the ability of simulation software to accurately represent complex physical processes.

Another limitation is the lack of software tools for gathering, processing and storing big manufacturing data. Digital twins require large amounts of data from various sources, including sensors, machines and other manufacturing systems. However, existing software

tools may not be capable of processing and storing these data in a way that allows for real-time analysis and decision-making. Despite these limitations, it is likely that advances in information technologies will enable Digital Twin based architectures to become more widespread in the future. For example, improvements in machine learning and artificial intelligence could help to improve the accuracy of simulation models and enable more efficient processing and analysis of manufacturing data. Additionally, advancements in cloud computing and data storage could help to address the challenges associated with managing and storing large amounts of data.

## 6. Conclusions

Cyber Physical Production Systems have become an important research topic as the enabler of the fourth industrial revolution. Using DT technology to improve real-time control, simulation and optimization of manufacturing operations is a matter of great importance. In this paper, we have improved the SUDIHA Supervised and Distributed Holonic architecture that we have proposed to dynamic scheduling for Cyber Production Systems. We used a DT Model to improve the Holons of this architecture. We have relied on the DT Model to deal with production system complexity. The DT-SUDIHA framework design consists of the Physical Objects PO (Product, Resource and Order) and the Virtual Objects VO configuration. The DT- SUDIHA operation is described by a diagram that follows part manufacturing and clarifies the interaction between Physical Objects and Virtual Objects of the shopfloor. This scheduling architecture allows handling, in real time, random disruptions. To show the validity of the proposed architecture, we have used it to improve the software tool used for the scheduling of the ENSAM flexible manufacturing cell. As a future research direction, we will explore its use of the Digital Twin in maintenance to develop Prognosis and Health Management, PHM, Systems and in Quality to develop Quality Advisory Systems, QAS. Another interesting future research direction we will explore is the use of blockchains to develop services that will be provided by the DT and used by Physical Resources.

**Author Contributions:** Conceptualization, H.K., S.S. and B.H.; methodology, H.K., S.S. and B.H.; investigation, H.K.; writing—original draft, H.K.; writing—review and editing, S.S.; supervision, S.S. and B.H. All authors have read and agreed to the published version of the manuscript.

**Funding:** This research received no external funding.

**Data Availability Statement:** The research data presented in this study are available on request from the corresponding author.

**Acknowledgments:** The use case was enabled by the High School of engineering ENSAM.

**Conflicts of Interest:** The authors declare no conflict of interest.

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
