# Peer review of "Digital Twin Based SUDIHA Architecture to Smart Shopfloor Scheduling"

_jmmp, doi:10.3390/jmmp7030084_

Round 1

Reviewer 1 Report

Dear Authors, 

Paper takes up an important and topical topic. The work meets the requirements of a scientific publication. However, I have a few comments, mainly to the drawings, which do not always reflect well the connections of the model presented in the publication, but in turn:

1. Abstract: it could be good to remove the definition of CPS from it and add more about the DT-SUDIHA Architecture model under study, what was new to this architecture, why it was created, what is its use 

2. Introduction, here you can give a definition of CPSs (note that in the article there is a singular CPS (system) or plural: CPSs), and even several authors and the structure of CPS according to Lee (https://doi.org/10.1016/j.mfglet.2014.12.001)

3rd About: Figure 1. Components of Cyber-Physical Machine Tool [11]

I do not understand the drawing why there is no connection between CNC machine tool and Machine Tool Cyber Twin.

Why the arrows are double-sided, and do not lead on the left side... and the right .., to nowhere

The drawing needs to be thought over and refined.

4. About: Figure 2: Five-dimension DT concept model. [20]

In my opinion, leganda under the drawing as a text record of the EP.... VE...., and then the drawing will be clearer (better)

5. About: Figure 3 : Proposed Multi Agent Scheduling Architecture [23]

The kanban arrow leads to what????????    It's about the left arrow..... Its direction has no purpose.

6. About: Figure 4 : SUDIHA Supervised Distributed Holonic Architecture

Kanban, in my opinion, you need links with the other elements of the HOLON model

7th About: Figure 6 : DT Based SUDIHA Framework

Why there is no connection between: smart product and smart resources (in my opinion it should be, due to LCA, for example)

8th About: Figure 7 : Structure of Resource Holon, font in the drawing too small

9. About: Figure 8 : Control flow of DT-SUDIHA Architecture, should simulation results not be in a yellow diamond

10. To the references, new papers could be added about DT and CPSs, and dyniamic planning (in my opinion)

Best wishes

Reviewer

Reviewer 2 Report

The paper aimed to present a digital twin to improve the Supervised and Distributed Holonic Architecture (SUDIHA) architecture proposed by the authors. A theoretical test scenario was presented. However, the results must be improved. 

The paper is organised. The recommendations are: 

1. I suggest defining the acronym only the first time it is used and after using it inside the text. It occurs in all the text. The authors need to revise it. For example, CPS was defined 5 times. Also, correct all typos when defining the acronyms. 

2.  Between lines 67 to 80, there are statements without references, and please add the references to the statements. It introduces important topics, such as Industry 4.0 and DT. For those who are reading is good to know where to find the content for a deeper study. 

3.  In lines 104, 105, and 106, “We will discuss, its use in manufacturing, achieve a state of the art about DT-based architectures used for scheduling and describe the five-dimension DT model which is a popular model used for DT design.” References, who used this?

4.   Correct the words in the text, use “Cyber-Physical” or “Cyber Physical”, choose one to be used in all the text, and correct “shopfloor” or “Shop floor” (e.g., line 393) to “shop floor”.

5.  Please increase the quality of Figure 2, Figure 3, Figure 7, and Figure 9 to meet the journal's requirements. At https://www.mdpi.com/journal/jmmp/instructions#figures

6.     In lines 261 and 262, please add the reference to the preview work. 

7. I suggest avoiding using ellipsis (…) and etc. If the reader is not from the field can miss important information. Also, for me, as a reader, it means that the authors are lazy to write. 

8.  Lines 487 and 488, “SUDIDA”, are correct, please.

9.  Line 533, “we used the RFID tag to Build the Product Twin”. Is it not a digital shadow? Once the RFID is used to read the information and, in this context, does not have bidirectional information, as the definition presented by the author between lines 113 to 125.

10. Lines 567 to 569, “DT-SUDIHA Architecture, implementation in today’s shop floors is a way to reach the goals of smart manufacturing systems. It improves the flexibility, efficiency, and autonomy of manufacturing operations. However, there are currently limitations to the implementation of this approach”. The authors must add proof for these statements, such as the time for communication, time for decision-making, latency during the communication, size of the database used to train the model, and more information. It is known by the rest of this paragraph that are difficulties. However, preliminary results should be provided.  

11.  English review.

12. Fill in the author’s contributions.

Reviewer 3 Report

The Digital Twin Arquitecture based SUDIHA is interesting and suitable for any reader in this field. However, some recommendations should be considered for publication:

1.     Method: consider include an outline about the research methodology carried out.

2.     Discussion: Authors should discuss the results and how they interpreted. Clarify this section, one option is creating a quantitative comparative study.

3.     How do you demonstrate the effectiveness of the proposed approach? Improve this question with a comparative or quantitative study.

Minor recommendations:

1.               Check references, there are mistakes. They do not adapt to the MDPI format. For example, the correct format of paper is: Author 1, A.B.; Author 2, C.D. Title of the article. Abbreviated Journal Name Year, Volume, page range.

Round 2

Reviewer 2 Report

The comments were addressed. There are no more comments from the reviewer.